# High Mannose N-Glycans Promote Migration of Bone-Marrow-Derived Mesenchymal Stromal Cells

**DOI:** 10.3390/ijms21197194

**Published:** 2020-09-29

**Authors:** Vivian Alonso-Garcia, Cutter Chaboya, Qiongyu Li, Bryan Le, Timothy J. Congleton, Jose Florez, Victoria Tran, Gang-Yu Liu, Wei Yao, Carlito B. Lebrilla, Fernando A. Fierro

**Affiliations:** 1Stem Cell Program and Gene Therapy Center, University of California Davis, Sacramento, CA 95817, USA; vivialonso@hotmail.com (V.A.-G.); ctchaboya@gmail.com (C.C.); bynle@ucdavis.edu (B.L.); timothy.congleton01@gmail.com (T.J.C.); jflorez@ucdavis.edu (J.F.); 2Institute of Biotechnology, Federal University of Uberlandia, Uberlandia, MG 38408-100, Brazil; 3Department of Chemistry, University of California Davis, Davis, CA 95616, USA; qyuli@ucdavis.edu (Q.L.); vktran@ucdavis.edu (V.T.); gyliu@ucdavis.edu (G.-Y.L.); cblebrilla@ucdavis.edu (C.B.L.).; 4Center for Musculoskeletal Health, Department of Internal Medicine, University of California Davis, Sacramento, CA 95817, USA; yao@ucdavis.edu; 5Department of Cell Biology and Human Anatomy, University of California Davis, Davis, CA 95616, USA

**Keywords:** mesenchymal stromal cells, amino-linked glycans, MAN1A1, Kifunensine, migration, bone fracture

## Abstract

For hundreds of indications, mesenchymal stromal cells (MSCs) have not achieved the expected therapeutic efficacy due to an inability of the cells to reach target tissues. We show that inducing high mannose N-glycans either chemically, using the mannosidase I inhibitor Kifunensine, or genetically, using an shRNA to silence the expression of mannosidase I A1 (MAN1A1), strongly increases the motility of MSCs. We show that treatment of MSCs with Kifunensine increases cell migration toward bone fracture sites after percutaneous injection, and toward lungs after intravenous injection. Mechanistically, high mannose N-glycans reduce the contact area of cells with its substrate. Silencing MAN1A1 also makes cells softer, suggesting that an increase of high mannose N-glycoforms may change the physical properties of the cell membrane. To determine if treatment with Kifunensine is feasible for future clinical studies, we used mass spectrometry to analyze the N-glycan profile of MSCs over time and demonstrate that the effect of Kifunensine is both transitory and at the expense of specific N-glycoforms, including fucosylations. Finally, we also investigated the effect of Kifunensine on cell proliferation, differentiation, and the secretion profile of MSCs. Our results support the notion of inducing high mannose N-glycans in MSCs in order to enhance their migration potential.

## 1. Introduction

Despite the hundreds of clinical trials demonstrating the safety of bone-marrow-derived mesenchymal stromal cells (MSCs; formerly designated as mesenchymal stem cells or multipotent stromal cells), robust clinical efficacy has only been achieved for a handful of indications [1,2]. It has been proposed that a major limitation has been the poor ability to target MSCs to tissues of interest [3]. MSC homing is inefficient, with only a small percentage of cells reaching the target tissue following systemic administration. This presents a major obstacle in realizing the full therapeutic potential of MSC-based therapies [3,4,5]. Therefore, it is imperative to improve the migration potential of MSCs in the interest of developing minimally invasive cell therapy.

The optimal delivery method for MSCs is often elusive. For example, clinical trials using MSCs to treat critical limb ischemia (CLI) have tested intravenous, intra-arterial, and intramuscular cell injections [6]. About half of all ongoing or completed clinical trials using MSCs use intravascular (either intravenous or intra-arterial) infusion as a route of delivery [7]. These methods are in general considered safe, although complications due to thrombosis and embolism have also been reported [7]. In a mouse model of CLI, we have shown that after intravenous injection, a small portion of MSCs is capable of homing to sites of hypoxia in the ischemic hind limb [8]. However, the vast majority of MSCs accumulates primarily in the lungs (and other filtering organs, such as liver and kidneys) due to steric hindrance [9,10]. Regardless of the route of delivery, targeting MSCs to specific tissues needs significant improvement.

Albeit much is known about genes involved in cell migration, the role of post-translational modifications is not well understood. Specifically, O- and N-linked glycosylations are common post-translational modifications that affect many functions including cell adhesion and signaling [11]. N-glycosylation are sugars attached to Asparagine in Asn-X-Ser/Thr motifs, where X is any amino acid except for proline. During the formation of these N-linked polysaccharides (N-glycans), a chitobiose core is attached to suitable asparagines in the ER, which are then highly decorated with mannoses to generate high mannose N-glycans. Subsequently, these mannoses are often trimmed by mannosidases (MAN1B1 in the ER and MAN1A1, MAN1A2 and MAN1C1 in the Golgi apparatus), allowing a plethora of other glycosyl-transferases to generate complex and hybrid N-glycans [12]. Therefore, a common notion is that high mannose N-glycans are precluded from other glycan modifications.

It has been proposed that N-glycans may affect cell migration [13], although the experimental evidence is mostly limited to selectin ligands (highly glycosylated molecules) and their role in cell tethering and rolling [3,4]. Since MSCs do not express selectins, the role of N-glycans on the migration of MSCs is largely unknown. However, it was shown that engineering N-glycans in MSCs by attaching antennary fucoses, induced homing to the bone marrow [14], supporting the notion that N-glycans are likely key regulators of MSC migration. In corroboration, we recently showed that basic fibroblast growth factor (bFGF or FGF2) promotes the migration of MSCs by increasing core-fucosylations in N-glycans, via transcriptional regulation of fucosyl-transferase 8 (FUT8) [15]. However, treatment with bFGF induces many alterations in MSCs, other than changes in N-glycoforms [16,17,18]. Some of these changes are likely undesired from a translational point of view (e.g., increased proliferation). Genetic manipulation of cells is generally associated with increased risks [19], while exo-core-fucosylation of N-glycans using recombinant FUT8 could be hampered by accessibility to the chitobiose and has not been demonstrated experimentally. Therefore, an ideal N-glycan signature for optimal MSC migration remains underdeveloped.

In cancer, high mannose N-glycans are associated with cell migration and metastasis [20,21]. In consequence, we hypothesized that high mannose N-glycans would also promote the migration of MSCs. Here we show that chemical or genetic inhibition of mannosidases leads to a greater migration potential of MSCs both in vitro and in vivo. As a potential underlying mechanism, we find that MSCs with high mannose N-glycans exhibit a smaller contact area with its substrates. In support of the chemical induction of high mannose for improved clinical use, we determined the temporal dynamics of these N-glycan modifications and the effect of Kifunensine on cellular functions, such as cell proliferation, differentiation and secretion of proangiogenic signals.

## 2. Results

### 2.1. Induction of High Mannose N-Glycans Promotes Migration of MSCs

To increase high mannose N-glycans in MSCs we used two approaches. First chemically, using the mannosidase I inhibitor Kifunensine [22,23]. Second, genetically by lentiviral transduction of cells with an shRNA targeting MAN1A1 (Figure A1). Although the chemical approach is practical for future clinical use, the genetic approach served as a complement to confirm the specificity of Kifunensine.

To test the effect of high mannose N-glycans on the migration of MSCs in vitro, we performed videomicroscopy, where the displacement of individual cells is tracked over time, and a wound/scratch assay, where migration of cells is assessed by means of closing a gap or “wound” in the monolayer of cells. Using videomicroscopy, treatment with either Kifunensine or shMAN1A1 caused a two-fold increase in cell migration, as compared to their respective controls (Appendix A and Figure 1A). These results were confirmed with the wound/scratch assays, where both Kifunensine and shMAN1A1 induced a significant increase in cell migration, as compared to their respective controls (Figure 1B).

To evaluate if nontreatment with Kifunensine would promote cell migration in vivo, we used two approaches. In the first model, immune-deficient mice underwent a controlled bone fracture in the diaphysis of one femur [15,24]. One hour after fracture, fluorescently labeled MSCs (pre-conditioned with Kifunensine or not) were injected intramuscularly, proximal to the fracture site. As shown in Figure 2A, three days after cell injection, MSCs treated with Kifunensine were more abundant in the muscle close to the fractured femur, as compared to control MSCs. These results were highly consistent (*n* = 4), but not quantifiable, because most inspected sections, regardless of treatment, did not show tdTomato^+^ cells. Together with our experiments in vitro, these results suggest that treatment with Kifunensine increases the active migration of cells.

In a second mouse model, immune-deficient NSG mice underwent ligation of a femoral artery, to induce hind limb ischemia [25,26]. One day after surgery, luciferase-expressing MSCs (treated as above) were injected via the tail vein. Cell distribution was then monitored based on luminescence. As shown in Figure 2B, at any of the measured time points, no significant number of cells could be detected preferentially in the ischemic limb. However, we noticed that treatment with Kifunensine increased the concentration of MSCs in the lungs both at one hour and 24 h after injection (Figure 2C). Since the lungs and other filtering organs are well known to be tissues where MSCs lodge following systemic administration (likely due to steric hindrance [10,14]), our results suggest that Kifunensine increases passive cell migration (i.e., cells being carried by the blood flow). To confirm that the effect of Kifunensine was on cell distribution and not on total cell numbers, we also measured total luminescence in the mice. We observed no major differences from controls, with exception of a small but significant increase of Kifunensine-treated cells 48 h after injection (Figure 2D). In consequence, these experiments suggest that although Kifunensine did not increase the tropism of MSCs toward areas of ischemia, it did favor the passive flow of the cells in the blood, possibly reducing the loss of cells in circulation.

### 2.2. The Effect of Kifunensine on N-Glycans of MSCs is Dynamic

We hypothesized that the effect of Kifunensine on N-glycans would be transitory and reversible. To precisely determine the dynamic changes of N-linked glycoforms (N-glycoforms) induced by Kifunensine, we used NanoLC/ESI-QTOF-MS (see Figure 3A for representative chromatograms). Here, we used two approaches. First, cells were cultivated for one day with or without Kifunensine. Then, the culture media was changed to standard culture media without Kifunensine. The cells were then cultured for an additional one and two days (Figure 3B). Following this approach and time points, the effect of Kifunensine was not reversed to basal levels. We also observed that the increase of high mannose correlated with a decrease of fucosylated N-glycans and complex or hybrid N-glycans. In contrast, the proportion of sialylated and both sialylated and fucosylated N-glycans did not change or was only minimally affected.

In a second experimental approach, cells were incubated for two days with or without Kifunensine. Then the culture media was changed to standard culture media without Kifunensine for an additional four days. Here, we found that after four days, the proportion of each N-glycoform was restored close to basal (control) levels (Figure 3C). Interestingly, exposure to Kifunensine for two days led to a higher percentage of high mannose N-glycans (83%), and consequently a stronger reduction of all other types of N-glycans, as compared to exposure to Kifunensine for only one day (with 60–70% high mannose N-glycans). These results suggest that the effect of Kifunensine is indeed reversible and that the increase of high mannose causes a reduction of fucosylated, sialylated, complex/hybrid N-glycans, and fucosylated/sialylated N-glycans.

### 2.3. High Mannose N-Glycans Induce Changes in Cell Shape and Stiffness

To elucidate how induction of high mannose N-glycoforms promotes cell migration we performed adhesion assays and assessed potential changes in cytoskeleton dynamics. In the adhesion assays, nontreated MSCs were let to attach over uncoated Petri dishes for 10 min. In these assays, Kifunensine had no visible effect on cell adhesion, while FGF2 (used as positive control), clearly reduced the attachment of cells to the dish (Figure A2).

To analyze cytoskeleton dynamics, MSCs were transduced to express the fusion protein LifeAct-tdTomato. LifeAct is a 17 amino acid peptide that binds to filamentous actin without interfering with actin dynamics [27]. We then performed videomicroscopy to assess changes in actin fibers over time but did not detect any marked differences in between controls and cells treated with Kifunensine, (not shown), Fierro, FA. Analysis of videos over various time resolutions recoding LifeAct-tdTomato expressing MSCs.

Although MSCs are characterized by a general fibroblastic morphology, cells are noticeably heterogenous in specific shapes and sizes. Some cells feature hallmarks of stressed or senescent cells, a morphology also commonly associated with fibrosis [28]. These cells are marked by abundant stress fibers (actin bundles), do not exhibit cell polarization, and are, therefore, rather polygonal and less spindle-shaped (Figure 4A). MSCs with these morphological characteristics exhibit very limited motility. Interestingly, treatment with Kifunensine reduced the frequency of cells with this “stressed” morphology (Figure 4B). However, since the overall percentage of cells with these features was very low (less than 2% of total cells), this is unlikely the cause for increased cell migration induced by Kifunensine.

In line with these observations, we observed that the contact area of cells to its substrate (i.e., the size of cells in two dimensions) correlates inversely with the speed of cells (Figure 4C). This is, the larger the contact area of a cell, the slower its motility. Interestingly, treatment with Kifunensine or the shRNA targeting MAN1A1 led to a strong reduction in contact area (Figure 4A,D), as compared to their respective controls, suggesting that the effective adhesion of MSCs to its substrate is reduced by the induction of high mannose N-glycans.

We next examined if the induction of high mannose N-glycans was affecting overall cell size. AFM measurements indicated that MSCs with shMAN1A1 are significantly taller as compared to MSCs with shControl. However, Kifunensine did not significantly affect cell height. To calculate total cell volume, we measured the radius of cells in suspension and found that the induction of high mannose N-glycans tended to increase cell volume, although differences were not significant (Figure 4F). These results suggest that the induction of high mannose N-glycans reduces the contact area of MSCs to the substrate without reducing overall cell sizes.

To assess the effect of high mannose N-glycans on single-cell mechanics, MSCs were categorized according to cell morphology to minimize cell to cell variations. We then focused on the dominant population, 33 ± 8 μm wide and longer than 80 μm MSCs transduced with either shControl or shMAN1A1. MSCs under both conditions exhibited similar morphological distribution. The mechanical properties, however, differed. As shown in Figure 4G, the mechanical profile of MSCs with shMAN1A1 (red) was right-shifted with respect to MSCs with shControl (blue), meaning that the genetically modified cells were softer or exhibit lower elastic compliance. Quantitatively, forces of 20.9 and 1385.4 nN were required to deform shControl cells by 30% and 80%, respectively. In contrast, a load of 4.6 and 645.53 nN was needed to deform shMAN1A1 cells by 30% and 80%, respectively. Using the initial portion of the profile (e.g., ε = 0–35%), the nonlinear least-squares fitting using the formula above yielded Young’s modulus: E_m_ = 4.1 and 0.8 MPa for the shControl and shMAN1A1, respectively. For all cells measured in three independent experiments, E_m_ equals 3.4 ± 1.9 Mpa for shControl cells versus 0.9 ± 0.7 Mpa for shMAN1A1 cells. These results suggest that silencing MAN1A1 made MSCs softer.

In contrast, when comparing MSCs treated with or without Kifunensine, the two profiles showed very similar mechanical compliance (Figure 4H). Loading forces of 6.6 and 779.4 nN were needed to deform control cells by 30% and 80% in height, respectively, while forces of 10.9 and 943.0 nN were required for Kifunensine-treated cells. The membrane’s Young’s modulus of the control and Kifunensine-treated cells is 0.8 and 1.4 Mpa, respectively. For all cells measured in the three replications, E_m_ equals 1.0 ± 0.6 and 1.2 ± 0.4 Mpa for control and Kifunensine-treated cells, respectively. These observations suggest that Kifunensine treatment did not seem to alter the elasticity of MSCs.

In conclusion, chemical or genetic induction of high mannose N-glycans alters the shape of MSCs by reducing the contact area to the substrate. Since reduced contact area correlates with higher cell motility (possibly due to reduced lodging to the substrate), our experiments suggest that the observed increase in cell migration is at least partially due to a change in cell shape. In addition, silencing genetically MAN1A1 makes MSCs softer, but this effect was not observed by treatment with Kifunensine.

### 2.4. The Migration-Inducing Effects of FGF2 and Kifunensine are not Additive

We recently showed that FGF2 increases the migration of MSCs by transcriptional changes in various genes associated with cell migration, but also on glycosyl-transferases. Specifically, FGF2 induces the upregulation of FUT8, which catalyzes the incorporation of core-fucosylations to N-glycans [15]. We therefore sought to determine if the pro-migratory effect of FGF2 and Kifunensine could be additive and if, therefore, we could achieve even higher cell migration than when stimulating with these factors separately. Using wound/scratch assays we found, as expected, that both FGF2 and Kifunensine increase cell migration. However, combining FGF2 and Kifunensine together did not yield any additive effect (Figure 5A).

A potential explanation for this lack of additive effect is that the high mannose N-glycans induced by Kifunensine could block FGF2 signaling. We therefore tested if migration-related genes regulated by FGF2 [15] were affected by preincubation of the cells with Kifunensine. As shown in Figure 5B, Kifunensine did not affect the regulation of genes by FGF2, suggesting that Kifunensine is not directly impairing FGF2 signaling.

A second explanation for the nonadditive effect of FGF2 and Kifunensine could be that high mannose N-glycans are precluded from core-fucosylations, because high mannose N-glycans in the Golgi Apparatus do not undergo further modifications. This notion is also supported by our mass spectrometry analysis (Figure 3), where the increase in high mannose N-glycans strongly correlated with a decrease in fucosylated N-glycans. To test this hypothesis, we transduced MSCs with either a control lentivirus or a lentivirus to overexpress FUT8. As previously reported [15], overexpression of FUT8 increased cell migration. However, ectopic expression of FUT8 in cells treated with Kifunensine did not increase migration. In fact, migration was reduced as compared to FUT8 or Kifunensine treatments separately, although differences were not statistically significant (Figure 5C). These results suggest that the migration-inducing effects of FGF2 and Kifunensine are possibly not additive because both factors compete for incompatible N-glycan modifications.

### 2.5. Kifunensine Affects Other Biological Processes of MSCs

Finally, we investigated if Kifunensine could affect proliferation, the secretion profile of angiogenic signals, or the osteogenic and adipogenic differentiation potential of the cells. To test the effect of Kifunensine on the secretion of angiogenic signals Angiopoietin 2, Interleukin 8 (IL-8) and Vascular Endothelial Growth Factor (VEGF-A), MSCs were treated for 24 h with or without Kifunensine, and then cultured for one additional day with or without Kifunensine. Angiogenic factors were then measured by ELISA. As shown in Figure 6A, treatment with Kifunensine induced a significant increase in secreted Angiopoietin 2 levels, but did not affect the secretion of IL-8 or VEGF-A. Cell proliferation was measured using MTT, where we found that cells cultured continuously with Kifunensine for up to six days, showed a significant increase in cell proliferation (Figure 6B). 

We then cultured MSCs for 14 days in either osteogenic or adipogenic media in the presence or absence of Kifunensine. In both differentiation assays, we found that continuous treatment with Kifunensine had a significant inhibitory effect, as assessed by Alizarin Red staining (Figure 6C, accounting for mineralization during osteogenesis) and Oil Red O staining (labeling lipid droplets within adipocytes; Figure 6D). However, pre-treatment for 24h with Kifunensine, did not affect the differentiation of MSCs. Altogether, these results suggest that sustained exposure to Kifunensine has a positive effect on cell proliferation, it has small or no effects on the secretion of proangiogenic signals and has an inhibitory effect on both osteogenesis and adipogenesis.

## 3. Discussion

The hereby conducted experiments support the notion of treating MSCs with Kifunensine to increase cell motility, which may favor the delivery of cells to target tissues. This approach is therefore likely to improve the therapeutic benefit of the cells for certain indications. The advantages of Kifunensine over other methods to promote migration of MSCs [3,4,5], include not having to modify the cells genetically and the minimal/no effects that pre-treatment with Kifunensine exerts over other cell functions.

Of note, most of the presented work focused on active cell migration, where cells are directly interacting with a substrate and displacing themselves based on the orchestration of cytoskeleton dynamics, adhesion molecules, and other factors. Clinical applications that require this type of migration of MSCs include those injecting the cells intramuscularly/percutaneously. Supporting this concept, MSCs treated with Kifunensine were more abundant in bone (especially close to the fracture site), as compared to controls. Although not examined, it is possible that at later time points (e.g., one or two weeks after injection), control MSCs will reach the fracture site too. However, since a high percentage of cells undergo apoptosis within days after injection, it may be critical to accelerate the delivery of cells to the site of damage, such as a fracture. Future studies are required to obtain quantitative data regarding active migration over time and to evaluate outcome measurements, such as bone repair.

In contrast to active migration, when cells are injected intravenously (as frequently used in clinical trials [7]), cell displacement is primarily driven by the blood flow (i.e., cell-extrinsic factors). The experiments using luciferase-expressing MSCs injected via the tail vein indicate that treatment with Kifunensine did not significantly increase the tropism of cells toward an area of ischemia. Specifically, we have previously shown that the sensitivity (i.e., signal over background) of this luciferase-based method is limited to 1000 cells, concentrated in a small volume [29]. Therefore, if cells were directed to the ischemic hind limb, it must have been in numbers below our detection threshold. However, we did observe consistent changes in the distribution of cells, where treatment with Kifunensine favored an increased concentration of MSCs in the lungs. We interpret these results as Kifunensine reducing the lodging of cells in blood vessels. From this, we speculate that the treatment may also favor intra-arterial delivery of the cells, although this remains to be empirically proven. Importantly, the mechanism for how high mannose N-glycans alter this passive migration remains unknown. We performed adhesion assays of MSCs over plastic (Figure A2) and over human umbilical vein endothelial cells (HUVECs; not shown) but did not observe any differences induced by Kifunensine. Recently, Regal-McDonald and colleagues showed that treatment of HUVECs with Kifunensine affects differently the adhesion of classical (CD14^+^CD16^-^) and nonclassical (CD14^+^CD16^+^) monocytes [30]. Those results suggest that changes in N-glycans affect diverse cells differently, depending on what glycoproteins are expressed. Other interactions to be studied include blood components such as red blood cells and platelets. In fact, the interaction of MSCs with platelets has emerged as critical for targeting MSCs to tissues of interest [31]. It is also possible that high mannose N-glycans cause changes in functional tissue factor (CD142), reducing coagulation. Despite the complexity of the potential interaction of MSCs with blood components, these remain critical to be fully understood for the advancement of effective MSC-based therapies that rely on intravascular cell delivery [7,9].

We found that the induction of high mannose N-glycans altered the shape of MSCs in vitro by reducing the contact area to the substrate, but not the total cell volume, as measured on the cells in suspension. Since we find a strong correlation between contact area and cell migration (i.e., the larger the contact area, the less the cells migrate), our results suggest that the induction of high mannose N-glycans promotes active cell migration by reducing the contact of cells to the substrate. The precise mechanism for this phenomenon remains unknown. As mentioned, we did not detect differences in cell adhesion, and in general, it remains largely unknown how changes in the plasma membrane, such as different N-glycoforms, may affect cell shape. One possibility is that changes in N-glycans alter the deformability of the plasma membrane, by altering its physical or biochemical properties [32]. In fact, we found that cells with shMAN1A1 exhibit a reduced elastic modulus, suggesting that these cells are softer. However, this effect was not observed in cells treated with Kifunensine, suggesting that the observed changes in cell stiffness are not the cause for reduced contact area and increased cell migration. In addition, it should be noted that even in MSCs transduced with shMAN1A1, other mannosidases are still expressed (e.g., MAN1A2, MAN1B1, etc.). Therefore differences in between the effect of Kifunensine supplementation and genetic silencing of MAN1A1 could be due to incomplete induction of high mannose N-glycans in the latter. Of note, it has been shown that Kifunensine increases the permeability of the plasma membrane in the enterocytic cell line Caco-2 [33]. Therefore, it is feasible that high mannose N-glycans induce other changes in the cell membrane, which consequently affect cell shape and migration. Finally, using molecular dynamics simulations we recently showed that high mannose N-glycans may affect the dimerization of transferrin receptor protein 1 (coded by the *tfrc* gene) in cholangiocarcinoma [21]. Since TFRC is also highly expressed in MSCs, it is also feasible that protein dynamics are altered by high mannose N-glycans, although this remains to be tested experimentally.

The studies on the dynamics of changes in N-glycoforms induced by Kifunensine led to interesting findings. First, even after Kifunensine is removed from the culture media, the increase of high mannose N-glycans remains for at least two more days. This could be due to sustained inhibition of type I mannosidases in the cell, but also due to the slow turnover of glycoproteins on the plasma membrane. Second, two days of exposure to Kifunensine (20 μg/mL) leads to a higher proportion of high mannose N-glycans, as compared to cells exposed to Kifunensine for only one day. This suggests that longer exposure to Kifunensine also leads to higher induction of high mannose N-glycans. Finally, it is interesting that the increase in high mannose N-glycans strongly correlated with reduced fucosylated N-glycans. In contrast, the proportion of sialylated N-glycans was only reduced upon very high increases in high mannose N-glycans.

In line with this inverse correlation of high mannose N-glycans and fucosylated N-glycans, we found that stimulation of MSCs with FGF2 (which induces core-fucosylations [15]) and Kifunensine did not yield to an additive effect on cell migration. One possibility for this is that Kifunensine could block FGF2 signaling. However, Kifunensine did not affect gene expression induced by FGF2 (including upregulation of FUT8). Since even overexpression of FUT8 could not induce migration in the presence of Kifunensine, our studies further support that the induction of core-fucosylations is a critical mechanism for how FGF2 induces migration of MSCs.

It was recently reported that incubation of an immortalized MSC line with Kifunensine (2 μg/mL) during osteogenesis caused an increase of calcium precipitation, but a reduction in gene expression of osteogenic markers [34]. In contrast, we found that Kifunensine inhibited mineralization. This discrepancy may be attributed to the concentrations of Kifunensine tested (20 μg/mL vs. 2 μg/mL), where perhaps a low concentration leads to only a small increase of high mannose N-glycans. Additionally, immortalized MSCs may inaccurately reflect the biology of primary cells (as used in our studies). Of note, the inhibitory effect of Kifunensine on both osteogenesis and adipogenesis was observed while incubating continuously the cells with Kifunensine during the entire differentiation period (14 days). Of note, it has been shown that cell shape is a strong determinant for differentiation [35,36], suggesting that the changes in cell shape caused by high mannose N-glycans may affect their differentiation. Finally, in further support of the use of Kifunensine, we saw a mild increase in cell proliferation and did not observe any negative effect on the secretion of angiogenic signals, suggesting that for many tested clinical applications, treatment with Kifunensine will not affect the functionality of the cells.

## 4. Materials and Methods

### 4.1. Isolation and Expansion of Human MSCs

MSCs were isolated from fresh bone marrow aspirates derived from healthy donors (both, male and female, 20–45 years old) (StemExpress, Folsom, CA, USA), based on the cells’ ability to attach to plastic [37]. Aspirates were mixed 1:1 with PBS, layered over Ficoll–Paque PLUS (GE Healthcare, Chicago, IL, USA) and centrifuged for 30 min at 600× *g*. Mononuclear cells were then plated in tissue culture flasks using standard culture medium. This is MEM-alpha (GE Healthcare) supplemented with 10% fetal bovine serum (Atlanta Biologicals, Flowery Branch, GA, USA) and 1% penicillin–streptomycin. After 2 days, nonadherent cells were washed off. In subsequent days, adherent cells expanded with the characteristic morphology, immune phenotype and trilineage differentiation potential of MSCs [38]. Experiments were performed with MSCs in Passages 2–7, where a passage represents 3 to 4 population doublings (5 to 7 days in culture). In our experimental conditions, this range of passages implies high purity of cells, and similar proliferation and differentiation rates, with no evident signs of senescence. Each experiment repetition (n) was performed with MSCs derived from a different donor. MSCs derived from different donors or at different passages were never pulled together. Treatment with Kifunensine (Santa Cruz Biotechnology, Dallas, TX, USA) was always with 20 μg/mL for 24 h, unless otherwise described.

### 4.2. Lentiviral Vectors and Transduction

Lentiviral constructs were cloned with the general form pCCLc-U6-shRNA-PGK-tdTomato, where shRNA is either a scrambled shRNA sequence that does not bind to any specific human mRNA target (shControl [18]) or an shRNA targeting MAN1A1, with the following targeting sequence: 5′-CATGAATTTGAAGAAGCAAAATCAT-3′. The vector with shControl was also used to label the cells with tdTomato in experiments transplanting cells into mice with bone fracture. The construct for the LifeAct lentiviral vector was obtained from Addgene, as generously deposited by Han [39]. To overexpress genes, constructs with the general form pCCLc-MNDU3-X-PGK-EGFP-WPRE were cloned, where X is either no sequence (control), the full-length protein-coding sequence of FUT8 (1728 bp) [15], or luciferase [26]. All transductions were performed using protamine sulfate (20 μg/mL) and with an amount of lentivirus equivalent to a multiplicity of infection (MOI) = 10, which is sufficient to generate 90–95% EGFP/tdTomato-positive MSCs, three days after transduction.

### 4.3. In Vitro Migration Assays

For wound/scratch assays, cells were seeded into 24-well plates with Cytoselect inserts (Cell Biolabs, San Diego, CA), at 50,000 cells per well (6 replicates per condition), and with or without Kifunensine (20 μg/mL) or FGF2 (10 ng/mL) for 24 h. The next day, inserts were lifted, and pictures were taken immediately and after 24 h. The wound area was quantified using TScratch software [40]. For experiments with engineered cells, transductions were performed 3 days prior to the assay.

For videomicroscopy, cells were plated in 35 mm Petri dishes (20,000 cells/dish), incubated as described, and placed in a BioStation microscope (Nikon, Tokyo, Japan), while maintaining the cells at 37 °C and 5% CO_2_. During recording, each dish was photographed every 5 min in 10 fields of view for over 20 h. Movies were analyzed using the plugin MTrack from ImageJ software to determine individual cell displacement over time (speed). At least 20 cells per MSC-donor per condition were analyzed.

### 4.4. In Vivo Migration Assay Toward Bone Fracture in Mice

All animal work was conducted strictly following institutionally approved animal protocols. Closed transverse diaphysis fractures of the right femur were generated at the mid-femur using a drop-weight blunt guillotine device, in 2-month-old mice as previously described [24], but in immune-deficient NOD/SCID IL2Rγ^–/–^ (NSG) mice. One hour after causing fracture, tdTomato-expressing human MSCs (treated with or without Kifunensine) were injected at 500,000 cells (resuspended in 20 μL PBS) per mouse, near the fracture site. This study was performed with four mice per condition. Two additional mice were injected with PBS only, to serve as negative control (to determine background fluorescence). After three days, mice were humanely euthanized, and samples fixed with formalin for 24 h, decalcified using 0.5 M EDTA (pH 8.0, USB Corporation, Cleveland, OH, USA) for an additional 24 h, and embedded in optimum cutting temperature (OCT) for cryosectioning. Cells were directly visualized based on tdTomato expression and DAPI staining for total nuclei. Sections were imaged using a BioRevo Keyence BZ-9000 fluorescence microscope (Keyence, Itasca, IL, USA) at ×10 for stitched images (low magnification) and at ×20 for high magnification images.

### 4.5. In Vivo Cell Migration under Hind Limb Ischemia and Luciferase Measurements

Unilateral hind limb artery ligation was performed as previously described [26]. In brief, NSG mice (5 mice per group) were anesthetized by inhalation of isoflurane, and skin cleaned with betadine and wiped with an alcohol pad. A 1 cm segment of the right femoral artery and all major collateral vessels were ligated using 5–0 monofilament suture (Moore Medical, Farmington, CT, USA) to induce complete hind limb ischemia. One day after surgery, luciferase-expressing MSCs (200,000 cells per mouse), treated with or without Kifunensine, were injected via the tail vein into mice. Luminescence was then measured using an IVIS system [25], one hour after injection and daily for the following three days.

### 4.6. Cell Shape and Cytoskeleton Dynamics

MSCs transduced with LifeAct were recorded using videomicroscopy as described above. From these images, we used ImageJ software to determine cell area (contact area), total perimeter, cell polarity, changes in cell shape over time, actin bundles/stress fibers and tail retraction, which is often considered the limiting step on fibroblast migration [41]. Large, flattened cells with cuboidal (not spindled) shapes with evident stress fibers were defined as “stressed” cells. These cells were counted under a fluorescent microscope in LifeAct-transduced cells treated with or without Kifunensine, over multiple fields of view (for a total of at least 300 cells per well). Cell counts were performed by three independent examiners.

### 4.7. Mechanics and Height Measurements at the Single-Cell Level

The mechanical properties and cellular height of living MSCs were measured at the single-cell level using atomic force microscopy (AFM), based on a method previously developed by our team [42,43]. The method, known as single-cell compression, allows determination of elastic compliance, adhesive force and cellular height measurement in one approach–retract cycle [42,44,45]. The detailed protocol has been previously reported [42]. Briefly here, single-cell compression was measured with an MFP-3D AFM (Asylum Research Corp, Santa Barbara, CA, USA) combined with an inverted optical microscope (IX80, Olympus America, San Jose, CA, USA). A 60× bright field oil-immersion objective (Olympus UPLFLN, San Jose, CA, USA) was used to allow precise positioning of the AFM probe above the nucleus of targeted cells and for direct visualization and monitoring of the cells during the experiments. The AFM cantilever (AC240, Olympus, San Jose, CA, USA) was modified by attaching a glass microsphere (60 µm diameter, Duke Scientific, Palo Alto, CA, USA) to the apex of the tip using a premixed two-component epoxy (Devcon, New York, NY, USA). The final spring constant of the modified probe was determined using the added-mass method [46]. The compression speed was 2 µm/s to avoid hydrodynamic effects. Force–deformation was acquired during approach and retract at two points: a clean surface area near the designated cell, and at the desired cellular contact. The probe-surface contact point during approach at the surface and at the cell allows the quantification of the cellular height. Force profiles during approach and retract were displayed as force versus relative deformation, ɛ, where ɛ = (h_0_ − h)/h_0_. h_0_ and h are the initial height and cellular height at the final compression, respectively.

Using Hertzian contact mechanics and assuming that cells behave like a balloon containing fluid, the force and relative deformation follow 3/2 of power law [42,43]:(1)$$F=\frac{\sqrt{2}E_{m}}{3\left(1-v_{m}{}^{2}\right)}R_{0}{}^{2}\varepsilon^{3/2},$$
where *F* is the load; *E_m_* represents the membrane Young’s modulus; Poisson ratio *ν_m_* = 1/2 for the cell membrane, i.e., it is incompressible; *R*_0_ is the radius of the uncompressed cell height; and ε is the relative deformation of cellular height (deformation/*R*_0_).

### 4.8. Gene Expression Analysis

RNA extraction was performed using a Direct-zol RNA Miniprep kit (Zymo Research, Irvine, CA, USA), following the manufacturer’s instructions. Real-time PCR was performed using TaqMan gene expression assays (Invitrogen, Carlsbad, CA, USA) and TaqMan Universal Master Mix reagents (Invitrogen), where the primers and probe are identified by the following Assay ID: MAN1A1: Hs00195458_m1, DOCK4: Hs00206807_m1, PODXL: Hs01574644_m1, RHOB: Hs05051455_s1*, FUT8: Hs00189535_m1, HMGA1: Hs00852949_g1*, HMGA2: Hs04397751_m1, FGFR2: Hs01552918_m1, GAPDH: Hs02786624_g1. For all assays, the probe spans over two exons, except for those with an asterisk (*), where both primers and probe map within a single exon.

### 4.9. N-Glycan Profile Analysis

One million MSCs treated for different time points with or without Kifunensine were lifted with Trypsin, washed once with PBS and resuspended in homogenization buffer (0.25 M sucrose, 20 mM HEPES-KOH (pH 7.4), and 1:100 protease inhibitor mixture (EMD Millipore, Burlington, MA)). Cells were then lysed on ice using a probe sonicator and lysates were pelleted by centrifugation at 2000× *g* for 10 min to remove the nuclear fraction and cells that did not lyse, followed by a series of ultracentrifugation steps at 200,000× *g* for 45 min to remove other nonmembrane subcellular fractions [47]. Membrane pellets were then suspended in 100 μL of 100 mM NH_4_HCO_3_ in 5 mM dithiothreitol and heated for 10 s at 100 °C to thermally denature the proteins. To release the glycans, 2 μL of peptide N-glycosidase F (New England Biolabs, Ipswich, MA, USA) was added to the samples, which were then incubated at 37 °C in a microwave reactor for 10 min at 20 watts. After the addition of 400 μL of ice-cold ethanol, samples were frozen for 1 h at −80 °C to precipitate deglycosylated proteins and centrifuged for 20 min at 21,130× *g*. The supernatant containing N-glycans was collected and dried.

Released N-glycans were purified by solid-phase extraction using porous graphitized carbon packed cartridges (Grace, Deerfield, IL, USA). Cartridges were first equilibrated with alternating washes of nanopure water and a solution of 80% (*v*/*v*) acetonitrile and 0.05% (*v*/*v*) trifluoroacetic acid in water. Samples were loaded onto the cartridge and washed with nanopure water at a flow rate of 1 mL/min to remove salts and buffer. N-Glycans were eluted with a solution of 40% (*v*/*v*) acetonitrile and 0.05% (*v*/*v*) trifluoroacetic acid in water and dried. The analysis was performed using nanoflow liquid chromatography/electrospray ionization quadrupole time-of-flight mass spectrometry (NanoLC/ESI-QTOF-MS) as previously described [47].

### 4.10. Western Blot

MSCs transduced with either shControl or shMAN1A1 were lysed for protein extraction using RIPA Buffer (Thermo Scientific) with 1% Halt Proteinase and Phosphatase Inhibitor Cocktail (Thermo Scientific). Proteins were extracted by strong agitation for 20 min at 4 °C, then centrifuged at 12,000× *g* for 10 min and stored at −80 °C. For western blots, 30 μg of proteins were loaded into 10% polyacrylamide gels and transferred into polyvinylidene fluoride membranes (BioRad). Blots were then incubated with an anti-MAN1A1 antibody (diluted 1:500, clone A13, Santa Cruz Biotechnology, Dallas, TX) overnight, followed by washes and incubation for 1 h with an HRP-conjugated anti-goat IgG secondary antibody (1:1000 Santa Cruz Biotechnology). After protein detection by chemiluminescence, membranes were stripped and incubated with an anti-beta-actin (diluted 1:1000, clone AC15, Sigma-Aldrich), followed by an HRP-conjugated anti-mouse IgG secondary antibody (1:1000, Santa Cruz Biotechnology) and detection by chemiluminescence.

### 4.11. Cell Adhesion Assay

MSCs were cultured for 24 h in standard culture media, with no supplement (control), FGF2 (10 ng/mL) or Kifunensine (20 μg/mL). Cells were then lifted using Trypsin, washed once with PBS and reseeded for 10 min in a 12-well plate (4.2 × 10^4^ cells/well in 4 replicates) over a rocking platform. Then, nonadherent cells are discarded, plates were washed twice with PBS, and remaining cells (i.e., attached cells) counted using Trypan Blue exclusion dye and hemocytometer.

### 4.12. Proliferation Assay

To determine the effect of Kifunensine on cell proliferation, MSC were plated in 96-well plates at 500 cells/well. At indicated time points, cells were incubated for 2 h in medium containing MTT dye solution (3-(4,5-dimethyl-2-Thiazolyl)-2,5-Diphenyl-2H-Tetrazolium bromide; Promega, Sunnyvale, CA, USA). Then, Solubilization/Stop solution (Promega) was added and incubated for one additional hour. Optical density was measured at 570 nm (with correction at 650 nm).

### 4.13. Secretion Profile Analysis

In order to detect secretion levels of Angiopoietin 2, Interleukin-8 (IL-8/CXCL-8) and Vascular Endothelial Growth Factor (VEGF-A), MSC (100,000 cells/well in 6-well plates) were cultured for 24 h with or without Kifunensine and then cultured for an additional 24 h with or without Kifunensine to collect the conditioned media. Then, supernatants (i.e., conditioned media) were collected and stored at −80 °C. Quantification of angiogenic factors was performed by enzyme-linked immunosorbent assay (ELISA) using the respective DuoSet kits (R&D Systems, Minneapolis, MN), following the manufacturer’s instructions.

### 4.14. Differentiation Assays

Osteogenic and adipogenic differentiation were performed as previously described [25,37], with an initial cell density of 10,000 MSCs/cm^2^ with regular medium changes every 3–4 days. Osteogenic medium was a standard culture medium supplemented with 0.2 mM ascorbic acid, 0.1 µM dexamethasone, and 20 mM β-glycerolphosphate. Adipogenic medium was a standard culture medium with 0.5 mM isobutylmethylxanthine, 50 µM indomethacin and 0.5 µM dexamethasone. Matrix mineralization was determined on Day 14 using the Alizarin Red S indicator (ARS; Ricca Chemicals, Arlington, TX, USA). Cells were fixed with 10% *v*/*v* formalin solution for 15 min, washed once with PBS, and stained for 20 min with 1% *w*/*v* ARS over gentle shaking. Samples were then washed with PBS, photographed for visual documentation. To quantify ARS staining, wells were incubated with 10% *v*/*v* acetic acid for 30 min, the cell layer scraped, vortexed for 30 s, and centrifuged at 12,000× *g* for 10 min. The optical density of the supernatants was measured at 405 nm.

For adipogenesis, cells were cultured for 14 days in adipogenic media. Then, cells were fixed with 10% *v*/*v* formalin solution for 15 min, washed once with PBS and stained for 30 min with Oil Red O (Electron Microscopy Sciences, Hatfield, PA, USA). Cells were then washed three times with PBS and incubated with 4% Tween 20 (Affymetrix, Santa Clara, CA, USA) in isopropanol for 5 min, in order to release the dye. The optical density of supernatants was then measured at 490 nm.

### 4.15. Statistical Analysis

Results are presented as mean with the standard error of the mean (SEM) as error bars. The number of biological replicates (experiments performed with MSCs derived from different donors) is indicated by “*n*” in each respective figure legend. Statistical differences were calculated using Student’s *t*-test, except for cell area, height and volume, which were first analyzed using a Shapiro–Wilk normality test. Since the values’ distribution was not Gaussian, the statistical significance of observed differences was calculated using a Mann–Whitney test. Correlation in between contact area and cell speed was calculated using Pearson’s correlation analysis. All statistical analysis was analyzed using Graph-Pad Prism Software. *p*-values < 0.05 were considered statistically significant.

## 5. Conclusions

Although preclinical studies testing the efficacy and safety of treating MSCs with Kifunensine are pending, our results support the notion of inducing a temporal increase of high mannose N-glycans, to promote both active and passive cell migration. This work illustrates how harnessing post-translational modifications is a viable strategy to alter positively properties of MSCs.

## Figures and Tables

**Figure 1 ijms-21-07194-f001:**
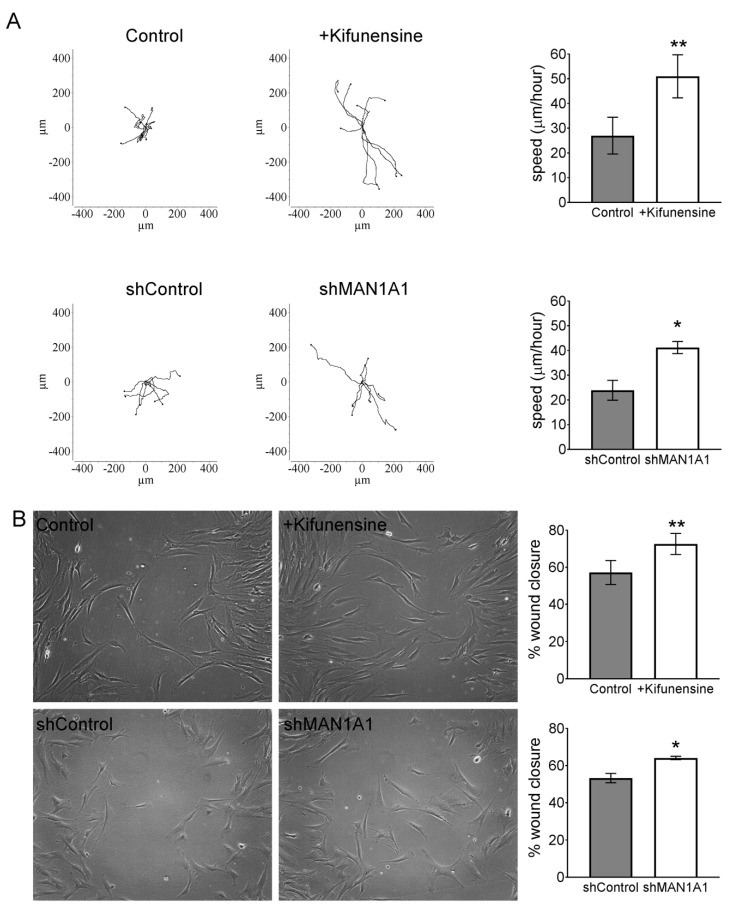
Inducing high mannose N-glycans increases the migration potential of mesenchymal stromal cells (MSCs) (**A**) Cell tracking after videomicroscopy. Trajectories of representative cells over 20 h are shown and quantified on bar graphs on the left (*n* = 3 for experiment with Kifunensine and *n* = 6 for experiment with shMAN1A1). (**B**) Wound/scratch assays. Representative images of wound closure after 24 h are shown, with quantification on bar graphs on right (*n* = 4 for experiment with Kifunensine and *n* = 5 for experiment with shMAN1A1). * *p* < 0.05 and ** *p* < 0.005.

**Figure 2 ijms-21-07194-f002:**
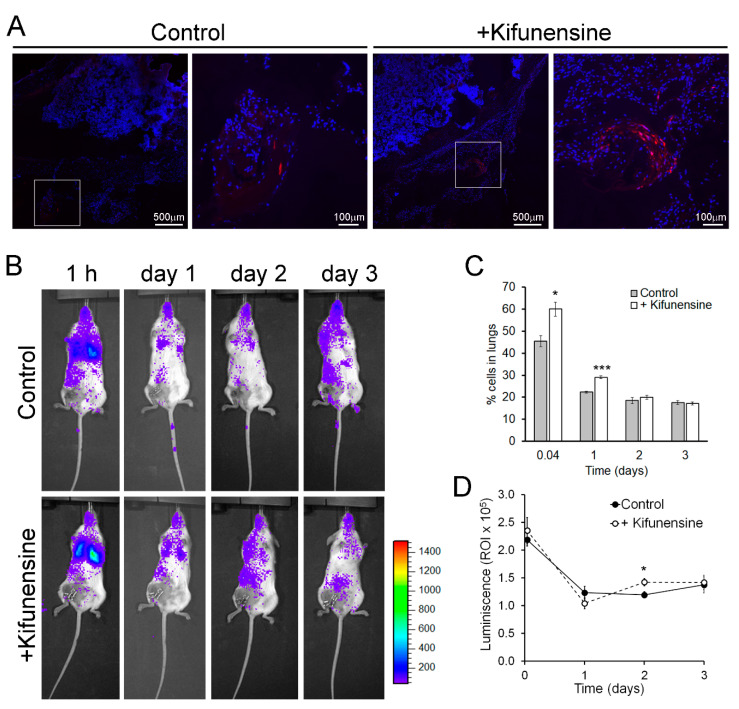
Pre-treatment with Kifunensine promotes the migration of MSCs in vivo. (**A**) Bone fracture model in NSG mice showing the distribution of tdTomato-positive MSCs (red) in the proximity of the fractured femur. High magnification images correspond to squares in low magnification images. Nuclei are stained with DAPI (blue). Cells (treated with or without Kifunensine) were injected percutaneously the same day of fracture and analyzed three days after (*n* = 4 mice per condition). (**B**) Mice with hind limb ischemia, where MSCs expressing luciferase were injected via the tail vein and imaged 1 h after or 1, 2 and 3 days after surgery. (**C**) Quantification of cells in the lungs (*n* = 5 mice per condition). (**D**) Total luminescence detected over time (*n* = 5 mice per condition). * *p* < 0.05, *** *p* < 0.0005.

**Figure 3 ijms-21-07194-f003:**
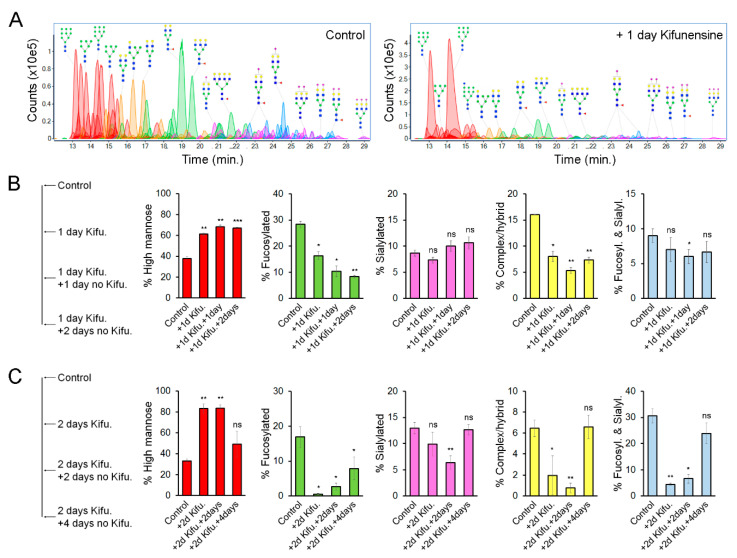
Effect of Kifunensine on N-glycans over time. (**A**) Representative chromatograms of N-glycans detected by MS of MSCs treated for one day with or without Kifunensine. (**B**) Semiquantification of N-glycoforms after 1 day of Kifunensine treatment and two additional days without Kifunensine. At these time points, increased high mannose N-glycans persist, while mostly fucosylated and complex/hybrid (not fucosylated and not sialylated) N-glycans remain low (*n* = 3). (**C**) Semiquantification of N-glycoforms in MSCs after 2 days with Kifunensine and up to four additional days without Kifunensine. Under these conditions, the increase of N-glycans correlated with strong reduction of other N-glycoforms, and four days after, Kifunensine treatment levels were restored to near control levels (*n* = 3). * *p* < 0.05, ** *p* < 0.005, *** *p* < 0.0005, *ns* = not significant.

**Figure 4 ijms-21-07194-f004:**
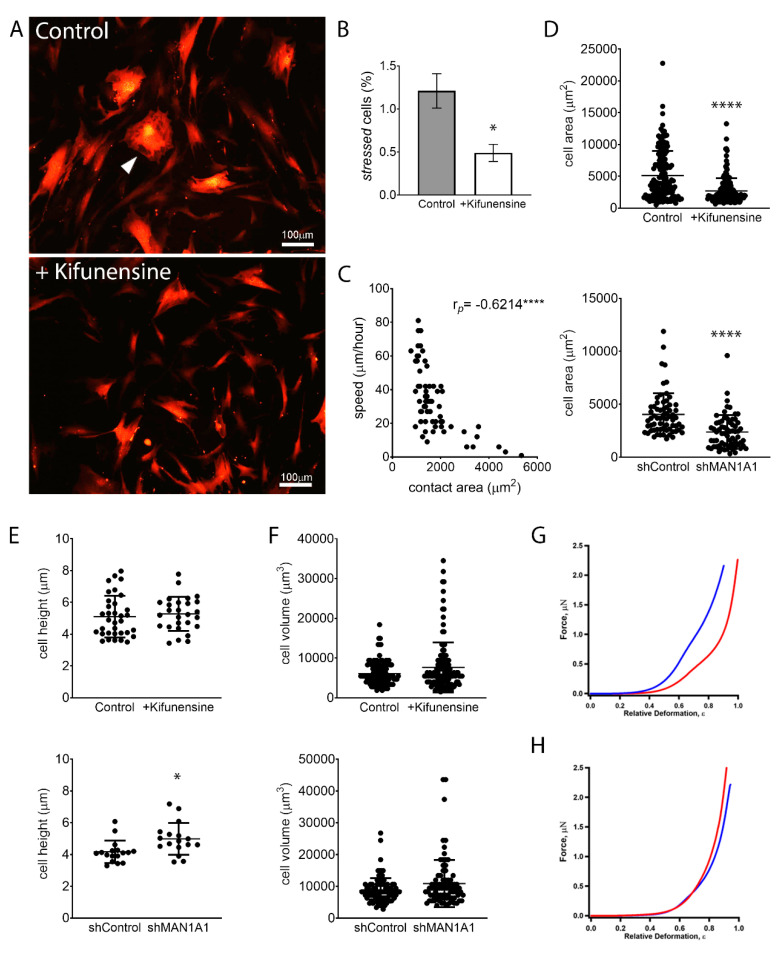
High mannose N-glycans promote changes in cell shape. (**A**) Representative images of MSCs expressing tdTomato, treated with or without Kifunensine. A “stressed” cell is indicated with the white arrowhead. Cells with Kifunensine also tend to look smaller. (**B**) Semiquantification of “stressed” cells (cells with morphological features as shown in A) (*n* = 4). (**C**) Correlation of cell size (contact area) and speed, as measured in control cells. Dots indicate individual cells and Pearson’s coefficient is shown in the upper right corner. (**D**) Contact area measured on cells visualized under a phase-contrast microscope. (**E**) Cell height measured by AFM. (**F**) Cell volume calculated by measuring cell radius on cells in suspension (under hemocytometer). For (**D**–**F**), *n* = 3 for both top and bottom graphs. (**G**) Force–deformation profiles of a representative shControl (blue) and shMAN1A1 (red) cell. (**H**) Force–deformation profiles of representative control (blue) versus Kifunensine-treated (red) cells. * *p* < 0.05, **** *p* < 0.00005.

**Figure 5 ijms-21-07194-f005:**
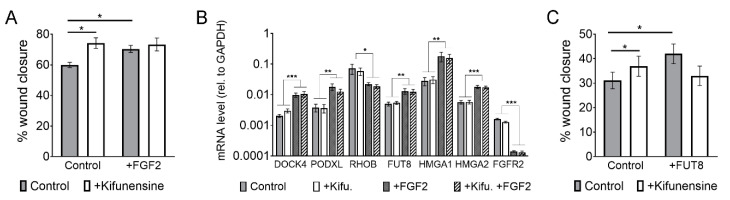
The effect of FGF2 and Kifunensine on migration is not additive. (**A**) Wound/scratch assays showing increased migration with FGF2 and Kifunensine, but no significant increase when both factors are combined (*n* = 3). (**B**) RT-PCR to genes regulated by FGF2, but not affected by Kifunensine (*n* = 5). (**C**) Wound/scratch assay of MSCs transduced with a control lentivirus or a lentivirus to overexpress FUT8 and treated with or without Kifunensine. Here, Kifunensine and overexpression of FUT8 promote migration, but no additive effect is observed when both conditions are combined (*n* = 4). * *p* < 0.05, ** *p* < 0.005, *** *p* < 0.0005.

**Figure 6 ijms-21-07194-f006:**
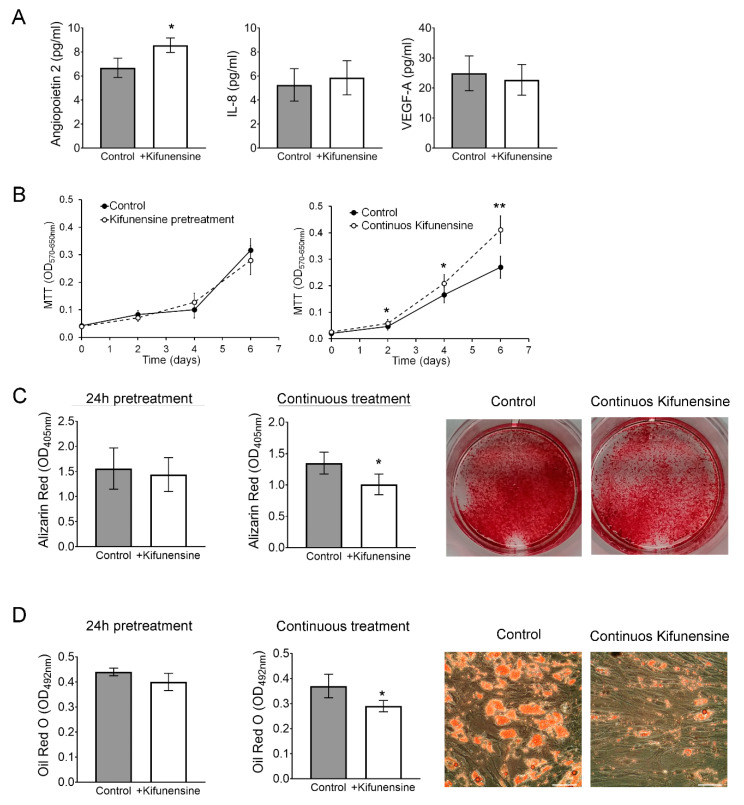
Continuous treatment with Kifunensine promotes migration, increases secretion of Angiopoietin 2 and inhibits both osteogenesis and adipogenesis. (**A**) ELISA to detect Angiopoietin 2, IL-8 and VEGF-A in the supernatant of MSCs treated for 24 h with or without Kifunensine (*n* = 6). (**B**) MTT assay on MSCs treated with or without Kifunensine (*n* = 6). (**C**) MSCs after 14 days in osteogenic media. Representative images of wells stained with Alizarin Red and quantification (*n* = 7). (**D**) MSCs after 14 days in adipogenic media. Representative images of Oil Red O staining and quantification (*n* = 5). * *p* < 0.05, ** *p* < 0.005. Scale bar = 100 μm.

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
