# Peer review of "High Mannose N-Glycans Promote Migration of Bone-Marrow-Derived Mesenchymal Stromal Cells"

_ijms, 2020, doi:10.3390/ijms21197194_

Round 1

Reviewer 1 Report

The Article is devoted to study of the approaches which will help to use the full therapeutic potential of mesenchymal stromal cells-based treatment due to improving mesenchymal stromal cells (MSCs) targeting to the tissue of interest which will promote the efficacy of MSCs application in tissue repair.

The research is carefully planned and professionally performed. The study includes consistently presented and logically described results of the experiments which are quite interesting and will allow to understand better the mechanism of MSCs migration and involvement of the high mannose N-glycans into this process. It is also quite important that glycans might possess lineage-specific nature and could probably serve as stem cells markers.

Minor points in the Article which do not diminish the value of the Article and refinement is recommended:

  1. Is there any opportunity that the process of transduction (using protamine sulfate) might influence cell mechanics? Probably it would be also interesting to compare Control and shControl groups?
  2. It is quite interesting to understand the mechanism of Kifunensine and FGF2 migration-inducing effects in connection with N-glycan modification. Which probably might be a good idea for further experiments.

Line 134 The bracket should be added at the end of the sentence: "experiment with shMAN1A1)."

Line 141 The sentence should be refined: “Mice with hind limb ischemia were MSCs expressing luciferase were injected via tail vein”

Figure 3B. The first “Control” column on the graph with the name "%Complex/hybrid" does not have error bars, probably it means that they are very close to the Mean?

Figure 3B. The name of the graph “%Focosyl. & Sialyl.” should be replaced by "% Fucosyl. & Sialyl."

Figure 6D. It would be better to add the scale bars to representative images of Oil-Red O staining.

Line 348 The following link “(39)” in the text should be replaced by “[39]”.

Line 388 It is required to specify the symbol at the end of the line: (2g/ml)

Line 564 The sentence should be refined: “To determine the effect of Kifunensine on proliferation of MSC were plated in 96-well plates at 500”

Small corrections should be done to the list of references according to the recommendations of International Journal of Molecular Sciences https://www.mdpi.com/journal/ijms/instructions#references.

Line 677 Should be refined as following: “biochemistry 2016, 117(9), 2128–2137.”

Line 692 Should be refined as following: “Induces Metabolic Changes, Enhances Survival and Promotes Cell Retention in Vivo. Stem Cells 2015, 33(6), 1818-28.”

Line 711 Should be refined as following: “impact on the function of resulting osteoblasts. Journal of cell science 2018, 131(4), jcs209452.”

Author Response

We thank Reviewer 1 for the positive feedback and the careful review of our manuscript. As detailed below, we hope to have addressed all concerns, suggestions and corrections. We are confident that these changes have improved the quality of our manuscript. Thank you very much.

Minor points in the Article which do not diminish the value of the Article and refinement is recommended:

1. Is there any opportunity that the process of transduction (using protamine sulfate) might influence cell mechanics? Probably it would be also interesting to compare Control and shControl groups?

It is correct that transduction per se could affect the cells. We therefore consider critical that the appropriate control for MSCs with shMAN1A1 are MSCs transduced with a control lentiviral vector (shControl). However, we did not examine the degree to how much transduction affects the cells by itself. All experiments with unmodified or transduced cells were conducted independently. They are therefore also often conducted with MSCs derived from different donors. This is the main reason why results of none-transduced MSCs (Control or Kifunensine) are not shown in the same graphs as transduced MSCs (shControl and shMAN1A1), as it would be misleading, since the experiments were not necessarily conducted side-by-side. We agree that comparing Control from shControl could be interesting, but we believe that plotting the data separately (i.e. exactly as the experiments were conducted) is more accurate and precise, as the effect of transduction itself is not directly the scientific questions we tried to address.

2. It is quite interesting to understand the mechanism of Kifunensine and FGF2 migration-inducing effects in connection with N-glycan modification. Which probably might be a good idea for further experiments.

We strongly agree with this comment. It is likely unnecessary for the “main point” of this manuscript, but worth to pursue in greater detail in the future.

We would like to express our gratitude to Reviewer 1 for the very careful suggested listed below. All corrections have been included to the revised version of the manuscript. The reviewer is correct in the assumptions that some error bars are very hard to see, because they are very small. Only the formatting of references is incomplete, due to incompatibility of using the journal’s template and the reference manager. We hope to incorporate those changes in the final proofs.

Reviewer 2 Report

This manuscript by Alonso-Garcia and co-workers describes an exploration of the effects of increasing the number of high mannose N-glycans on mesenchymal stromal cells (MSCs). The authors accomplish this by two methods: treatment of the cells with kifunensine, a small molecule inhibitor of alpha 1-2 mannosidases like mannosidase alpha class 1A member 1 (MAN1A1); or by silencing MAN1A1 using shRNA. They find that pre-treatment of MSCs with either kifunensine or anti-MAN1A1 shRNA promotes cell migration in vitro. In vivo models yield results consistent with the in vitro findings, although the MSCs do not necessarily migrate to lesions. They show that treatment with kifunensine suppresses processing of MSC N-glycans in the Golgi, leading to an increased abundance of high mannose N-glycans, and the effect is reversible. Finally, they present evidence that cells with an overrepresentation of high mannose N-glycans have smaller contact areas with surfaces, which enables them to migrate faster, and that when they are derived by treatment with anti-MAN1A1 shRNA, the cells are more easily deformed.

This manuscript is easy to follow and the results are interesting. Therefore, I believe it would be appropriate for publication in IJMS after the authors address a few comments as follows:

  1. While it seems clear that knocking down MAN1A1 with shRNA should result in an increase in high mannose N-glycans much like treatment with kifunensine, the authors do not show this directly like they do with kifunensine. If possible, the authors should do an experiment like the one shown in Figure 3 for the MAN1A1 shRNA knockdown cells. The results could be somewhat different from those obtained with kifunensine, given that there are other mannosidases in the cis-Golgi, like MAN1A2, that have redundant activity with MAN1A1. Knockdown of MAN1A1 with shRNA would be specific only for MAN1A1 whereas kifunensine inhibits all alpha 1-2 mannosidases.
  2. The authors state that knockdown of MAN1A1 results in the cells becoming softer. Consistent with this conclusion, the Young’s modulus is 3.4 MPa for shControl cells and 0.9 MPa for shMAN1A1 cells. However, the same experiment yielded Young’s moduli of 1.0 and 1.2 MPa for cells with and without treatment with kifunensine. The discrepancy between the controls in these experiments is troubling: 3.4 MPa for shControl cells and 1.0 for non-kifunensine-treated controls. This suggests that the difference between shControl and shMAN1A1 cells may be due to an idiosyncrasy in the measurement on the shControl cells rather than a treatment effect.

Author Response

1. While it seems clear that knocking down MAN1A1 with shRNA should result in an increase in high mannose N-glycans much like treatment with kifunensine, the authors do not show this directly like they do with kifunensine. If possible, the authors should do an experiment like the one shown in Figure 3 for the MAN1A1 shRNA knockdown cells. The results could be somewhat different from those obtained with kifunensine, given that there are other mannosidases in the cis-Golgi, like MAN1A2, that have redundant activity with MAN1A1. Knockdown of MAN1A1 with shRNA would be specific only for MAN1A1 whereas kifunensine inhibits all alpha 1-2 mannosidases.

We appreciate this commentary and respectfully request to not conduct this additional experiment, as it would take several months. Please notice that the purpose of Figure 3 is not to show the efficacy of Kifunensine to induce high mannose N-glycans, but to study the dynamics (i.e. the effect over time). Since expression of shMAN1A1 is permanent (i.e. viral constructs are incorporated into the cell’s DNA) we do not expect changes over time. The studies on shMAN1A1 are also meant to confirm our findings with Kifunensine. Nevertheless, Reviewer 2 is correct in noticing that other mannosidases are likely still expressed. We have incorporated such comment into the discussion.

2. The authors state that knockdown of MAN1A1 results in the cells becoming softer. Consistent with this conclusion, the Young’s modulus is 3.4 MPa for shControl cells and 0.9 MPa for shMAN1A1 cells. However, the same experiment yielded Young’s moduli of 1.0 and 1.2 MPa for cells with and without treatment with kifunensine. The discrepancy between the controls in these experiments is troubling: 3.4 MPa for shControl cells and 1.0 for non-kifunensine-treated controls. This suggests that the difference between shControl and shMAN1A1 cells may be due to an idiosyncrasy in the measurement on the shControl cells rather than a treatment effect.

We thank Reviewer 2 for this comment. We hope to have partially clarified this with the abovementioned text included in the discussion. It also should be noted that all experiments with unmodified or transduced cells were conducted independently. They are therefore often conducted with MSCs derived from different donors. This is the main reason why results of none-transduced MSCs (Control or Kifunensine) are not shown in the same graphs as transduced MSCs (shControl and shMAN1A1), as it would be misleading, since the experiments were not necessarily conducted side-by-side. We agree that comparing Control from shControl could be interesting, but we believe that plotting the data separately (i.e. exactly as the experiments were conducted) is more accurate and precise, since the effect of transduction itself is not directly the scientific questions we tried to address.